# Bioprocess of Gibberellic Acid by *Fusarium fujikuroi*: The Challenge of Regulation, Raw Materials, and Product Yields

**DOI:** 10.3390/jof10060418

**Published:** 2024-06-12

**Authors:** Aranza Hernández Rodríguez, Adrián Díaz Pacheco, Shirlley Elizabeth Martínez Tolibia, Yazmin Melendez Xicohtencatl, Sulem Yali Granados Balbuena, Víctor Eric López y López

**Affiliations:** 1Centro de Investigación en Biotecnología Aplicada, Instituto Politécnico Nacional, Carretera Estatal Santa Inés Tecuexcomax-Tepetitla, Km 1.5, Tepetitla de Lardizábal, Tlaxcala 90700, Mexico; ahernandezr2201@alumno.ipn.mx (A.H.R.); ymelendezx2000@alumno.ipn.mx (Y.M.X.); 2Unidad Profesional Interdisciplinaria de Ingeniería Campus Tlaxcala, Instituto Politécnico Nacional, Guillermo Valle, Tlaxcala 90000, Mexico; adiazpa@ipn.mx (A.D.P.); sgranadosb@ipn.mx (S.Y.G.B.); 3Instituto de Investigaciones en Materiales, Universidad Nacional Autónoma de México, Mexico City 04510, Mexico; smartinezt@materiales.unam.mx

**Keywords:** plant growth regulator, bakanae, fermentative processes, production yields, gibberellins market

## Abstract

Gibberellic acid (GA_3_) is a tetracyclic diterpenoid carboxylic acid synthesized by the secondary metabolism of *Fusarium fujikuroi*. This phytohormone is widely studied due to the advantages it offers as a plant growth regulator, such as growth stimulation, senescence delay, flowering induction, increased fruit size, and defense against abiotic or biotic stress, which improve the quality and yield of crops. Therefore, GA_3_ has been considered as an innovative strategy to improve agricultural production. However, the yields obtained at large scale are insufficient for the current market demand. This low productivity is attributed to the lack of adequate parameters to optimize the fermentation process, as well as the complexity of its regulation. Therefore, this article describes the latest advances for potentializing the GA_3_ production process, including an analysis of its origins from crops, the benefits of its application, the related biosynthetic metabolism, the maximum yields achieved from production processes, and their association with genetic engineering techniques for GA_3_ producers. This work provides a new perspective on the critical points of the production process, in order to overcome the limits surrounding this modern line of bioengineering.

## 1. Introduction

Abiotic stress, comprising drought, heavy metals, salinity, and temperature variations, is one of the biggest problems facing the agricultural sector worldwide, because it negatively affects phenotypic characteristics, physio-biochemical processes, and crop plants production, causing large economic losses and low crop yields. Despite this, the evolution of plants has focused on the development of biochemical and molecular mechanisms that help defend them from these alterations [1,2].

The exogenous application of plant growth regulators (PGRs) is currently one of the most studied defense tools. Although PGRs are synthesized innately by plant species, their supply at low concentrations has shown a significant improvement in plant growth and resistance to abiotic or biotic stresses. PGRs can be defined as natural or synthetic compounds that specifically interfere with the development or metabolic processes of higher plants (natural hormonal system), without contributing to their nutritional or phytotoxic value [3]. Figure 1 shows the most notable group of PGRs, composed of the auxins (IAA), abscisic acid (ABA), cytokinins (CTKs), brassinosteroids (BRs), jasmonates (JAs), salicylic acid (SA), ethylene (ETH), and gibberellins (GAs) families, the last representing compounds of high agro-industrial interest [4], as will be explained in the next sections.

GAs are known as a large family of tetracyclic diterpenoid carboxylic acids, whose specific members (GA_1_, GA_3_, GA_4_ and GA_7_) have functional bioactivities that provide the role of growth hormones in plants. Gibberellic acid (GA_3_), produced by *Fusarium fujikuroi*, is the major phytohormone commodity employed in the horticultural, viticultural, agricultural, and downstream brewing industries due to its role in growth stimulation, developmental process triggering, seed germination, stem elongation, sex expression, flowering, fruit formation, and senescence [5,6]. These features make GA_3_ an effective and beneficial strategy to improve the quality and productivity of agri-food interest crops, leaving aside the harmful consequences for the environment.

Although the highest yields of GA_3_ (1442.85 mg/kg/day and 487.50 mg/L/day) were, respectively, obtained by solid-state and submerged fermentation of *F. fujikuroi* [7,8], the production levels currently achieved are not representative to cover the demand of the global market. The size of the gibberellin market has been estimated at USD 1.32 billion in 2024, and is expected to grow at a compound annual growth rate (CAGR) of 7.56% to reach USD 2.06 billion by 2030 [9]. However, the lack of production strategies with parameters that boost the biosynthesis of the phytohormone at a large scale significantly limits its application and commercialization. Unfortunately, the GA_3_ market is limited exclusively to first world countries, when it could represent an imminent opportunity for developing countries in response to the requirements for high quality agricultural products. In this context, there is an increasing need to search and innovate methodologies that allow the optimization of the GA_3_ production process to achieve higher yields.

Considering GA_3_ as a promising strategy for the benefit of the agricultural sector, the aim of this article is to, particularly, review the origin and nature of the phytohormone in question, the biosynthetic pathway, its effects on crop plants, and its role in resistance against abiotic stress. Moreover, the optimization of different production processes (with an emphasis on raw materials) was discussed, and the use of native and genetically modified strains is compared according to the reported yields. The aim is opening new insights that englobe all required variables for the establishment of the most accurate production parameters that maximize the GA_3_ productivity to enhance the world market, a crucial component of the green revolution.

## 2. Bakanae: From the Problem to the Opportunity

The *fungi* kingdom has attracted attention for its incredible role in the innate biosynthesis of various metabolites, highlighting one of the most important gibberellins (GA_3_) for the manufacturing industry. Although large-scale production and phytohormone application at low concentrations is currently a crucial point for the development of modern and efficient agricultural practices, excess GA_3_ in plant material can cause adverse responses such as the hypertrophy, rotting, and blackening of roots, chlorosis, etiolation, rotting of the crown, sterility (panicles without grains), lesions on the leaf surface, the development of adventitious roots, delayed growth, and even premature death [10].

Therefore, the discovery of the GA_3_ molecule was not exactly beneficial for the 19th century when the rice crop (*Oryza sativa* L.) suffered a severe fungal infection that significantly reduced its quality and yield. This plant disease was called bakanae, which means “foolish” or “bad” seedlings in Japanese, in reference to the effects on crops caused by the infection, with the abnormal elongation of seedlings as the characteristic symptom (induced by the surplus of GA_3_ secreted by the fungus after infection of the host) [11,12]. The disease was initially attributed to *Fusarium heterosporum* Nees [13], but the name of the pathogen was re-established as *Gibberella fujikuroi* Sawada [14]. Consequently, the reproductive stages were defined in *Gibberella fujikuroi* Sawada as the teleomorph (sexual reproductive stage) and in *Fusarium moniliforme* Sheldon [15] as the anamorph (asexual reproductive stage) of the fungus [12]. However, the causal organism was eventually particularized and renamed as *Fusarium fujikuroi* Nirenberg [13], the teleomorphic state of *G. fujikuroi* Sawada [12]. From the previous knowledge, the understanding of disease epidemiology is presented in Figure 2, which constitutes a key point to reduce qualitative and quantitative losses, at least, in the main rice-producing countries whose yield losses reach up to 50% in Japan, 3.0–95% in India, 6.7–58% in Pakistan, 40% in Nepal, and 28.8% in Korea [16].

Bakanae is a monocyclic infection transmitted mainly by seeds from the previous growing season infected with ascospores, but it also survives in plant remains and soils in the form of macroconidia or thick-walled hyphae. Although the viability of these is reduced over time under field conditions [10], the survival rate of the pathogen can vary from 4 to 10 months [17]. When infected seeds are propagated in the next growing season, infected seedlings show symptoms between 5 to 10 days post-inoculation. Generally, although the seedlings do not present symptoms at the time of transplant, those infected wilt before the heading stage or, failing that, continue their development invaded by white or pinkish mycelium on the plant surface, which makes evident the levels of disease severity. In severe cases of infection, the plant usually wilts 10 to 40 days post-inoculation. However, the conidia formed in the tillering stage are dispersed by climatic conditions such as rain or wind during the flowering stage, infecting the floral organs and, therefore, the seeds of later stages whose glume, embryo, or endosperm are invaded. Thus, seeds whose surface is colonized can continue to act as primary inoculum in the next planting, where most of the seedlings wilt during germination, but if they show a greater development, it is usually late and with a sterile panicle emergence stage [18].

Despite the epidemiological history of *F. fujikuroi*, the discovery of GA_3_ biosynthesis potential has been highly useful for scientific-technological research, showing its ability for the overproduction of GA_3_, which has positioned it as a biofriendly alternative with prospects for application at the industrial level, providing a new source of obtaining a metabolite with beneficial properties on agri-food crops. Since GA_3_ regulates more than just vegetables stature and seed dormancy, it additionally impacts plant features such as cambial activation, stem elogation, tissue expansion, flowering, and floral development, directly or indirectly. As presented, its outstanding contribution to increased yields and efficiency of crops defines it as a crucial component of the green revolution [19].

## 3. *Fusarium fujikuroi* as the Main GA_3_ Producer

In spite of the advances made regarding the pathogen and the bakanae disease cycle, current research has indicated that several *Fusarium* species are associated with infection development. Although *F. fujikuroi*, *F. proliferatum*, *F. verticillioides*, and *F. andiyazi* belong to the *Fusarium fujikuroi* species complex (FFSC), and that they have been identified as being responsible for causing bakanae disease [20], *F. fujikuroi* (belonging to genus *Fusarium* of Ascomycota) has the highest degree of infection and it is the only one that can synthesize GA_3_. It causes counterproductive symptoms in crops such as corn, cotton, sugarcane, and rice [12,21], from approximately 50 phylogenetic species grouped in the American, African, and Asian clades of the FFSC complex [22].

While the isolates of *F. proliferatum* and *F. verticillioides* can vary in their pathogenicity depending on the crop [12], the virulence of *F. fujikuroi* in the susceptible rice genotype Pusa Basmati 1121 showed a disease severity higher than 70%, compared to that shown by *F. proliferatum* (43%) and *F. verticillioides* (62%) [23]. Furthermore, it has been reported that *F. fujikuroi* presents the highest frequency of isolates collected from infected rice plants with symptomatic bakanae, with an incidence of 80.05%, followed by *F. proliferatum* with 8.31% and *F. andiyazi* with 0.95% [24], which makes it the most dominant species among the *Fusarium* spp. associated with bakanae disease in rice cultivation [25].

Morphologically, *F. fujikuroi* can produce pale orange sporodochia, frequently hidden by mycelium and chains of microconidia. Macroconidia are slender, lengthy, and have three to five septa without much curvature. Microconidia have an oval or club-like shape, a flattened base, at least one septate, and are frequently found in chains of polyphialids or fake heads. Nonetheless, *F. fujikuroi* shares a lot of morphological similarities with *F. proliferatum* and other species from the FFSC complex. For instance, comparing their colony morphologies, it is almost indistinguishable from *F. proliferatum* and shares characteristics with *F. verticillioides*; the only methods for accurate identification are DNA sequencing or cross-sexual fertility tests. Despite this, *F. fujikuroi* owns unique genes that have been cloned and studied; the most prominent are associated with the regulation of nitrogen, carbon, carotenoids, and gibberellic acid. Additionally, some strains of *Fusarium* carry the gene encoding either functional trichothecene 3-O-acetyltransferase or fumonisin production. Moreover, *F. fujikuroi* has gene clusters that encode polyketide synthase (PKS), which produces the red pigment bikaverin. However, a different pigmentation can be observed based on the culture conditions; for instance, in solid medium *F. fujikuroi* can show a different range of pigments, including grayish orange, violet gray, dark violet, and dark magenta, or even display no pigmentation [26,27,28].

## 4. Gibberellic Acid (GA_3_)

Gibberellic acid (GA_3_) has a different application in the agro-industrial field, and it is easier to obtain compared to the other phytohormones; however, it is still obtained in minimum quantities. GA_3_ is a tetracyclic diterpenoid compound that contains double bond of differing reactivities (C1–C2), an axial secondary alcohol, a tertiary alcohol, a γ-lactone ring (C10), and a carboxyl group (C7). In 1982, E. J. Corey synthesized GA_3_, but it required several manipulation steps due to the high density of its functional groups. It has been reported that GA_3_ can also be produced by plants and certain microorganisms including bacteria and fungi. Figure 3 shows the complexity of the molecule, with a graphical representation of its associated benefits and its synthesis by fungi. Geranyl-geranyl diphosphate (GGPP), a typical C20 precursor for diterpenoids, is usually the first step in the biosynthesis process [29,30,31].

GA_3_ is a white crystalline powder that melts between 233 and 235 °C, and it can be dissolved in alcohol, acetone, ethyl acetate, and butyl acetate. Although it is stable under dry conditions, it is not the most stable of the GAs compounds, since it quickly decomposes at high temperatures, alkaline pH levels, and also in aqueous solutions, with a limited solubility in benzene and chloroform. Furthermore, the loss of the γ-lactone ring represents the biological inactivation of GA_3_ [32].

## 5. Effects of GA_3_ on Crops

Gibberellic acid promotes physiological and metabolic responses in plants, when interacting with climatic factors, such as temperature, solar radiation, and humidity. It has been reported that the application of GA_3_ at ppm levels (parts per million, mg/L) can lead to the elimination of dormancy in seeds, the acceleration of germination, fruit development, and the overcoming of dwarfism [33]. In particular, commercial products such as ACTIVOL^®^ 40% GS (from Valent BioSciences^®^ LLC., Libertyville, IL, USA) recommend applying doses between 4 and 100 ppm of the active ingredient (GA_3_) in volumes of water ranging from 100 to 4000 L (depending on the type of crop). As an example, the correct product dose per hectare for foliar application over rice crops is 6–12 g, when the seedling consists of two to five true leaves. This benefits the increase, vigor, acceleration, and uniformity of seedling development, allowing the emergence time and flood irrigation amount to be brought forward (7–14 days), maximizing the crop yield. On the other hand, when the treatment is applied on seeds before sowing, the dose is 2.5 g/50 kg of seed, which helps the development of underground stems, the emergence of the seedling, and the density and uniformity of the established crop [34]. In the case of other products such as BIOGIB^®^ 10PS (from Arysta LifeScience^®^, Cary, NC, USA), the dose and time of application for strawberry crops, for example, is 10 g per 100 L of water at the beginning of fruit growth, with a second dose between 15 and 20 days later, in order to increase the fruit size [35].

Despite the advantages that industrialized products offer to agricultural fields, it is important to emphasize that the natural GA_3_ molecule intrinsically stimulates plant growth mechanisms, such as foliar gas exchange (photosynthesis, stomatal conductance, and transpiration rate), sink formation, cell division and expansion, higher leaf nutrient concentration (nitrogen, potassium, and phosphorus), and the development of xylem, root, stem, shoots, and leaf area [36]. Likewise, it induces plant transition processes, such as growth from the meristem to the shoot, from the juvenile to the adult leaf stage, and from vegetative growth to flowering [37]. Figure 4 presents some of the principal biological activities of GA_3_.

From another point of view, GA_3_ plays a crucial role in regulating abiotic stress. It has been studied that the exogenous application of this PGR suppresses the negative effects of salinity stress, as it reduces the concentration of sodium (Na^+^) and hydrogen peroxide (H_2_O_2_) in shoots and roots, increases the dry and fresh weight of plant material, improves total chlorophyll content, and increases antioxidant gene expression and antioxidant enzymatic activity [38]. GA_3_ can also improve tolerance to drought stress by attenuating the adverse effects on cell membrane stability, ion leakage, relative content and efficient use of water, photosynthesis-related foliar pigments (chlorophylls and carotenoids), photosystem II efficiency (PSII in terms of Fv/Fm and performance index), mineral nutrient content, lipid peroxidation, antioxidant capacity, and osmoprotectants [39]. Likewise, GA_3_ is involved in the adaptation mechanisms of plants against stress caused by heavy metals such as lead (Pb) and cadmium (Cd) because it improves photosynthetic and mitotic activity (cell division rates), increases the net rate of CO_2_ assimilation, improves chlorophyll and proline accumulation, and increases the activity of enzymes involved in the elimination of reactive oxygen species (ROS). Furthermore, it increases the content of endogenous amino acids, which can influence the regulation of membrane permeability, osmolyte accumulation, and ion absorption, thereby increasing tolerance to metal stress [40].

Finally, it has been reported that endogenous GA_3_ regulates the resistance to cold stress, maintaining the integrity of cell membranes during low temperature storage, reducing the accumulation of superoxide anion (O_2_^−^), improving antioxidant capacity, and activating the feedback mechanism of gibberellin anabolism and the expression of C-Repeat Binding Factor1 (CBF1), an important regulator of cold resistance, thus effectively improving the cold resistance of fruits [41].

For all these reasons, it has been considered that the exogenous application of the phytohormone plays an indispensable role in the post-harvest life of horticultural crops and ornamental plants, since GA_3_ is closely related to the ability of delaying fruit senescence. This is because it promotes a significant reduction in the accumulation of malonaldehyde (MDA) and H_2_O_2_, improves catalase (CAT) and superoxide dismutase (SOD) activities, and reduces the peroxidase (POD) and polyphenol oxidase (PPO) activities in shoots, effectively slowing down the aging process [42]. Furthermore, treatment with gibberellins reduces the intensity of respiration, increases pulp hardness, inhibits the release of endogenous ethylene, and represses fruit softening and ripening. It also improves the storage quality of fruits by modulating their shape, size, flavor, color (the content of pigments such as chlorophylls, carotenoids, and anthocyanins), and nutritional value, in addition to delaying the reduction of soluble solids and titratable acids at harvest, promoting the accumulation of sugar, and avoiding the loss of vitamin C, phenols, and soluble proteins, thus maintaining the life and quality of post-harvest vegetables [43].

Based on the research reported to date, it is magnificent how a PGR such GA_3_ can be able to influence the development of plants and stimulate the necessary armament for them to survive and succeed against biotic and abiotic stresses. Leaving sublime perspectives on the development of an elementary, economic, and innovative production system, GA_3_ is prospected as a promising tool for sustainable agriculture, where crop yields are increased without altering the nature of their production.

## 6. GA_3_ Biosynthesis Regulation

Since their discovery, GAs’ biosynthesis has been exhaustively studied by many research groups, highlighting the great interest in understanding how these metabolites are codified, regulated, and expressed in fungus and also in plants. For this revision, we place special emphasis on GA_3_ produced in *Fusarium fujikuroi*, where the biosynthetic pathway is a complex process that involves several steps for completion. Multiple regulators play a role in the activation or repression of enzymes along the whole process. Since GAs are secondary metabolites, their biosynthetic pathway is influenced by the primary metabolism, where the glycolysis and Krebs cycle pathways provide the precursors (carbon/nitrogen sources) needed for hormone biosynthesis, and this also can modulate the regulatory activities for production [44,45]. Figure 5 shows this interconnection, suggesting that these previous steps must be always considered for stablishing an efficient bioprocess for GA_3_ production.

From the start, it is known that the codifying region comprises a biosynthetic gene cluster that includes a desaturase (des) gene, four cytochrome P-450 monooxygenase genes (*P450-1*, *P450-2*, *P450-3*, and *P450-4*), and a geranylgeranyl diphosphate (GGDP) synthase region (*ggs2*), which act together with the (*cps/ks*) complex involved in the early steps of regulation and expression of the first steps for ent-kaurene production [46].

The sequential steps for GA_3_ production have been well described by many authors; hence, a brief description and diagram is given in this section. Figure 5 corresponds to the principal pathway of GA_3_ biosynthesis, from which the interaction of the components from the gene cluster is observed, as previously mentioned, starting from the farnesyl diphosphate (FDP) compound. Then, the production of the intermediate GGDP is achieved by the synthase activity from GGS to give ent-copalyl diphosphate (CDP) and then ent-kaurene, to obtain the resultant compound ent-7a-hydroxykaurenoic acid [47]. Afterwards, multifunctional P450 monooxygenases play a central role in the metabolism of intermediate molecules, catalyzing several steps from the further process.

Finally, to obtain the end-product GA_3_, the activity of these multifunctional monooxygenases has an important intervention for the oxidation and hydroxylation steps [48,49,50], which must be tightly regulated.

According to the normal growth conditions, it has been reported that secondary metabolism from fungi is modified as a function of different environmental factors like temperature, light, pH, and carbon and nitrogen sources [51]. Regarding nitrogen regulation, this has been extensively studied by many research groups, where the main findings were that GA_3_ biosynthesis is repressed by the presence of nitrogen in a high amount (from different sources) [52]. AreA is one of the major transcriptional factors that modulates both the activation of GA_3_ biosynthetic genes under nitrogen limitation [47] and the de-repression of genes involved in nitrogen metabolism [53]. Moreover, it has been observed that AreA can influence the expression of six genes from the GA_3_ biosynthetic cluster (*des*, *P450-4*, *P450-1*, *P450-2*, *ggs-2*, *cps/ks*) by the direct binding to their promoters at the GATA/TATC regions [53]. Accordingly, the regulation of AreA has been also extensively studied, showing the interplay between AreA and other regulators such as Nmr, a repressor of AreA activity that acts by protein–protein interaction [54], and MeaB, which maintains a relationship with AreA that governs the regulation of bikaverin biosynthetic genes [55]. Moreover, AreB is another transcription factor that can act either as an activator or a repressor of genes under the control of AreA, binding also to GATA/TATC elements from gene promoters, and contributing to the positive regulation of GA_3_ production under nitrogen starvation [56].

However, it is known that fungi and bacteria can adapt their metabolisms depending on the nutrients available in the environment. Therefore, the regulatory systems can activate different inducers or repressors for the control of carbon catabolite and nitrogen metabolic repression [57]. This is important since preferred sources such as glucose (carbon), ammonium, and glutamine (nitrogen) will always be preferably used and will delay the use of alternative carbon and nitrogen sources.

Hence, it is important to establish the regulation as well of GA_3_ biosynthesis by carbon source conditions, which has been inferred but not completely elucidated for the metabolism of *F. fujikuroi*. For instance, Rios-Iribe et al. (2011) determined that the production of gibberellins can be affected by the use of glucose as the only carbon source, since it promotes fast growth by switching off secondary metabolites. They found that a mixture of carbon sources can promote the slow assimilation but increased production of GA_3_, demonstrating the influence exerted by the carbon metabolism [58]. Regarding carbon regulation, it has been described in *Aspergillus nidulans* that the repressing regulatory activities of AreA and AreB depend on the carbon source, particularly about the repression of arginine metabolism in the presence of glucose for AreA and of fructose for the case of AreB [57].

In summary, the understanding of the biosynthetic pathways involved in GA_3_ production is quite important for its improvement [5], and must be considered for the establishment of bioprocess production processes to satisfy agro-industrial needs.

## 7. GA_3_ Production

At a large scale, the main process for producing GA_3_ has been developed through fermentation using different biological models, with *Fusarium moniliforme*, *Fusarium proliferatum*, *Fusarium fujikuroi*, and *Paecilomyces* as the main producers [32,59]. In addition, there are also reports of production with *Aspergillus niger* [60,61,62], *Penicillium variable* [63], *Inonotus hipidus* [64], *Pleurotus eryngii* [65], and *Fusarium oxysporum* [66]. However, the yields reported with these fungi are low. The three main production modes in bioreactors are semi-submerged (SSSF), solid-state (SSF), and submerged (SmF) fermentation; nevertheless, GA_3_ production is currently carried out mainly by SSF and SmF [32,59]. Table 1 summarizes the main studies that have focused on GA_3_ production, and includes information about substrates, operating conditions, daily maximum yields, and cultivation strategies.

Regarding substrate, the first studies determined that slowly utilized carbon sources benefited GAs production [67]. Since then, the composition of the media has considered the use of cost-effective carbon sources such as flours, pulps, seeds, pods, bran, husks, and bagasses, which also allow to maintain a nitrogen deficit [33]. This available information has allowed that different works can be focused on the use of raw materials with the purpose of solving two problems, the increase of GA_3_ production and the waste utilization of residues from different agro-industries. The most employed raw materials are starch [68], corn flour [69], wheat bran [70], rice bran [7], barley flour, citrus pulp [71], corn starch [72], rice flour [8], and soybean flour [73], among others.

Many research works have suggested a variety of strategies to optimize production, with the most common being the optimization of growth parameters such as incubation time, cultivation temperature, pH, humidity, and water activity (the last two for SSF), as is analyzed in Table 1. On the other hand, further research suggests a broader range of approaches, including coupling extraction during fermentation [74,75], the use of mutant strains [69,72], models and simulations [76,77], pH and temperature regulation [61,65], and the testing of different carbon–nitrogen ratios (C:N) [8,78]. Next, we discuss the relationship between the GA_3_ concentration reached when using raw materials and the incubation conditions; the most interesting reports of GA_3_ production are described from a process point of view.

In the case of SSF, there are three reports that we highlighted regarding the obtained concentration of GA_3_. Qian et al. [70] reported 1750 mg/kg/day, employing a strain of *Fusarium moniliforme* M-7121 in a substrate consisting of 20% starch and 80% maize flour/wheat bran. The bioreactor was a 3 L drum, operated for 12 days at 28 °C, with an initial humidity of 60%, a pH of 5, and a water activity ranging from 0.98 to 0.99%. It is worth noting that the authors declared the utilization of a low carbon–nitrogen ratio. In other work, the second highest yield (1468 mg/kg/day) was reported by De Oliveira et al. [71], employing, also, a *Fusarium moniliforme* strain LPB03 using citric pulp as the substrate. The initial operating conditions comprised 75% relative humidity, a temperature of 29 °C, and a pH ranging from 5.5 to 5.8 for five days in a 0.25 L column bioreactor (the authors employed a “high” C:N ratio). The strain *Gibberella fujikuroi* NRRL 2278 produced the third-highest yield at 1442.85 mg/kg/day [7]. The medium consisted of unprocessed rice bran, barley malt residue, and glucose, maintained for seven days at a temperature of 28 °C and 70% relative humidity in a 400 mL flask.

The three studies agree on maintaining an initial humidity between 60 and 75%, an initial pH between 5 and 5.8, and a temperature of 28–29 °C for 5 to 12 days. The major difference is the use of different substrates and cultured strains. However, *Fusarium moniliforme* exhibited the highest yields across all experiments. Another notable point is the cultivation system, with the highest reported yield per day achieved using a rotating drum bioreactor (1750 mg/kg/day). This contributed to maintaining an improved mass transfer compared to the packed columns (1468 mg/kg/day) or flasks (1442.85 mg/kg/day) used in other reports.

On the other hand, SmF fermentation has been the main method used to produce GA_3_ at industrial level. Within this strategy, it can be highlighted that the most notable yield reached a final concentration of 15 g/L using 30 g/L of sucrose, as reported by Rangaswamy [79], with a daily production yield of 1500 mg/L/day using the strain *Fusarium moniliforme* NCIM 1100, optimizing the physiological parameters at a temperature of 30 °C, pH of 7.0, and agitation (150 rpm). Although the authors did not describe the C–N relation and the product/substrate yield (YP/S), they were calculated giving a C:N ratio of 15.5 (considering NaNO_3_ as the nitrogen source), and YP/S = 0.5, with a total solid concentration (TSC) in media of 35 g/L. This information is in contrast to the common asseveration about the high C:N relation near of 100 that must be maintained to reach a high GA_3_ production.

Another interesting report conducted by Escamilla et al. [8] achieved a yield of 487.50 mg/L/day in a fluidized bioreactor using mycelium of *Gibberella fujikuroi* Sawada (CDBB H-984) immobilized with Ca-polygalacturonate. The evaluation of the operational conditions indicated that glucose (100 g/L) and rice flour (2 g/L) used as carbon sources, and NH_4_Cl (1.0 g/L) used as nitrogen source and operated at a temperature of 30 °C, pH of 5.0, and aeration of 3 vvm for 8 days, resulted in a two-fold increase in phytohormone production. Considering the provided data about the 3.9 g/L of GA_3_ obtained and the initial glucose concentration reported by the authors, a YP/S of 0.043 was calculated. The production medium had a TSC of 107.5, which means that the medium components were not only utilized for gibberellic acid biosynthesis.

The two studies agree on cultivation conditions between 28 and 30 °C and between 5 and 7 pH being optimal. Similarly to SSF, the best daily yield was obtained with the *Fusarium moniliforme* strain, followed by the *Gibberella fujikuroi* strain. Production times do not differ much compared to SSF, reaching their maximum yields per day between 7 and 10 days. Similarly, reports with higher yields employ easily assimilable substrates such as glucose, sucrose, and rice flour, unlike the third study, which uses a complex mixture of agro-industrial residues. Despite many research efforts, *Fusarium moniliforme* is well known for the production of mycotoxins like fumonisin B1 and B2 [80,81]. Fumonosin B1 has been reported as a tumor promoter in the development of esophageal cancer [82]. This may be a point to consider when using GA_3_ in crops intended for human consumption.
jof-10-00418-t001_Table 1Table 1Gibberellic acid (GA_3_) production by fermentation.MicroorganismSubstrateCultivation StrategyOperation ConditionsProduction (g/L) Productivity (g/L/day)Ref.*G. fujikuroi*CDBB H-984Rice flour 2 g/LGlucose 100 g/LFluidized immobilized myceliumOptimization temperature, pH and C:N ratioBatch fluidized bioreactor (3.5 L)C:N 36.8pH 530 °C3 vvm 8 daysAround 3900 mg/L487.50 mg/L/day[8]*G. fujikuroi*NRRL 2278Glucose 40 g/L30% raw rice bran/70% barley malt residueSSF ^2^Agro-industrial wastes from rice processing and brewing industryBatch flask (50 g in 400 mL)28 °CpH 5–5.570% moisture7 days10,100 mg/kg1442.85 mg/kg/day[7]*F. moniliforme*LPB03Citric pulp 30 gDifferent fermentation systemsSSSF ^1^, SSF ^2^ and SmF ^3^Column bioreactor (30 g in 0.25 L)75% moisture30 mL/min airpH 5.5–5.829 °C5 daysSSF ^2^ results:7340 mg/kg1468 mg/kg/day[71]*F. moniliforme*M-7121Starch maize flour 800 gWheat bran 200 gSSF ^2^Optimization moisture, water activityBatch drum reactors (3 L)60% initial moisture28 °CInitial pH 512 days21,000 mg/kg1750 mg/kg/day[70]*F. moniliforme*NCIM 1100SmF ^3^: sucrose, 30 g/LSSF ^2^: jatropha seed cake, 5 gSSF ^2^ and SmF ^3^Batch flask SmF ^3^:30 °C10 days150 rpmMonitored 1 dayBatcha flask SSF ^2^:30–45 °C10 daysMonitored 2 daysSmF ^3^:15,000 mg/L1500 mg/L/daySSF ^2^:10,500 mg/kg1050 mg/kg/day[79]*G. fujikuroi* DMS 893Glucose 120 g/L Glycine 4 g/LExtractive SmF ^3^
Genapol 2822Fed-batch Bioreactor (6 L) 29 °C 0.8 vvm 400 rpm 20.83 days 1050 mg/L 50.40 mg/L/day[74]*G. fujikuroi*NRRL 5538Cassava flourSSF ^2^Different support (cassava flour, sugar cane bagasse and low density polyurethane)Column fermenter (20 g cassava) 29 °C 1.5 days 250 mg/kg 166.67 mg/kg/day[83]*G. fujikuroi*NRRL 2284Glucose 80 g/LSmF ^3^Effect of C:N ratio 7.97–796.8 Batch Fermenter (1.8 in 3 L) C:N 122 pH 5 700 rpm 30 °C 1 vvm 7.08 days 1000 mg/L 141.24 mg/L/day[78]*G. fujikuroi*CDBB H-984Glucose 50 g/LSmF ^3^
Kinetic model simulationAirlift bioreactor (3.5 L) pH 3 29 °C 1.6 vvm Around 12 days Around 100 ppm 8.3 ppm/day[76]*G. fujikuroi*CDBB H-984Glucose-corn oilSmF ^3^Mixture of carbon sourceBatch bioreactor (7 L)pH 3.529 °C600 rpm1 vvm12 days380 mg/L31.67 mg/L/day[58]*G. fujikuroi*LPB 06Citrus pulpSoybean hulls (5%)SmF ^3^Agroindustrial wastes, citrus pulp and soybean hullsStirred tank reactor (2 L)29 °C0.5 vvm500 rpm4 days205.2 mg/L51.3 mg/L/day[84]*G. fujikuroi*CDBB H-984GlucoseCorn oilSmF ^3^Mixture carbon sourceBatch bioreactor (7 L)pH 3.529 °C600 rpm1 vvm15 days430 mg/L28.3 mg/L/day[85]*F. moniliforme*LPB06Coffee husk 10 gCassava bagasseSSF ^2^Strain selectionBatch flask(10 g in 250 mL)pH 5.370% moisture29 °C7 days492.5 mg/kg70.36 mg/kg/day[86]*Aspergillus niger*Czapek-Dox brothSmF ^3^Optimization of incubation time, temperature, pH, agitation.Batch flask (100 mL in 250 mL)30 °CpH 5150 rpm12 days238.7 mg/L19.89 mg/L/day[61]*Aspergillus niger*FüRSANMolassesSmF ^3^Food industry wastes (molasses, vinasse, whey, suger-beet)Batch flask30 °C12 days150 rpm155 mg/L12.92 mg/L/day[62]*Penicillium variable*Banana peelSucrose 2%SSF ^2^Olive oil as natural precursorStatic flask reactor27 °CpH 57 days31.95 mg/kg4.56 mg/kg/day[63]*Inonotus hipidus*Sucrose 30 gSmF ^3^Optimization, immobilization and kinetic parametersBatch flask4% inoculant30 °C150 rpm21 days2990 mg/L142.38 mg/L/day[64]*Pleurotus eryngii*Fructose 50 g/LSmF ^3^Optimization, incubation, temperature, pH, and agitationBatch flask25 °C150 rpmpH 718 days7920 mg/L440 mg/L/day[65]*Fusarium**oxysporum*Sesame brakSucrose 3 gSSF ^2^Medium components and culture conditionsBatch flask(10 g in 500 mL)30 °C8–10 days8160 mg/kg816 mg/kg/day[66]^1^ Semi-submerged fermentation (SSSF), ^2^ solid-state fermentation (SSF), and ^3^ submerged fermentation (SmF).

## 8. GA_3_ Production Improvement by Genetic Engineering and Bioprocess Optimization

As observed in the previous section, GA_3_ production has been obtained from the bioprocess with different fungal strains, and the strategies for production enhancement have been mainly focused on fermentation media optimization. However, from the last 15 years, the current proposals have considered production improvement by using genetic engineering methods. This section will be focused on the recent advances made using *Fusarium moniliforme* and *Fusarium fujikuroi* and the enhanced production obtained by genetic modifications, presented in Table 2.

For instance, the first attempts considered the development of random mutations acquired by irradiation from different sources, with subsequent screenings to find out improved mutant strains, selected according to their morphological characteristics of size, sporulation metabolism, and pigmentation [87,88]. Afterwards, the generated knowledge about regulatory circuits for GA_3_ biosynthesis opened new opportunities to perform mutations and genetic modifications, enabling for the carrying out of profound analyses for explaining the effects observed on metabolism. It was from these works that the strong relation was established between the effects on production by the obtained mutants and the corresponding optimization of media cultures, from which GA_3_ production was improved 28.6% during flask cultures and a 33.3% increment was obtained when cultured at bioreactor, compared to the original strain [69]. As shown, the generated knowledge about regulatory circuits for GA_3_ biosynthesis opened new opportunities to perform mutations and genetic modifications, enabling researchers to carry out profound analyses for explaining the effects observed on metabolism.

This is the case for the work developed by Zhang et al. (2020), where mutation improvement in GA_3_ producers was explained by genome sequencing, finding that genes involved in endocytosis, the MAPK signaling pathway, and histone modifications were responsible for the increased production of GA_3_ [72].

Currently, recent proposals have established that global engineering modifications could be even more useful for increasing product yields. For instance, Wang and co-workers (2022) suggested a multivariate strategy of metabolic engineering, where three modules from the GA_3_ biosynthetic pathway consisting of a precursor pool, a cluster-specific channel, and a P450-mediated oxidation were optimized to achieve a reconstitution of the metabolic balance of the industrial GA_3_ producer strain, which showed an increased production of 48.2% [89]. Likewise, the most recent report presents a regulatory modification strategy based on the overexpression of AreA and Lae1, two regulators that positively influence GA_3_ biosynthesis [90]. From this work, the third-highest production rate was achieved, of 431 mg/L/day, by coupling promoter engineering, metabolic modification, and transcriptome analysis in the strain *Fusarium fujikuroi* CGMCC No: M 2,019,378. Bidirectional co-expression of the genes geranylgeranyl pyrophosphate synthase 2 (Ggs2) and cytochrome P450-3 showed an improvement in GA_3_ production. The cultivation conditions were 28 °C, 600 rpm, and an external air flow rate of 10 L/min. According to the authors, its principal components were medium corn starch, rice flour, soybean meal, and peanut powder, and mineral salts were given at 178.61 g/L of TSC. This transformed strain reached 5.48 g/L and 2.89 g/L GA_3_ of biomass and GA_3_, respectively, over a 7-day fermentation. The C:N ratio calculated was 48.1. Despite the fact that the authors augmented the GA_3_ compared to that of the native strain, it seems that the raw materials are not directed towards to GA_3_ synthesis.

As presented, engineered strains have opened new perspectives to increase GA_3_ biosynthesis, which requires good complementation with bioprocess optimization. The complexity of the GA_3_ biosynthetic pathway suggests that further studies must improve regulatory circuits at the global level, manipulating, simultaneously, both genetic and metabolic elements.
jof-10-00418-t002_Table 2Table 2Genetic engineering strategies for gibberellic acid (GA_3_) improvement.Micro-OrganismGenetic Modification StrategyCultivation ModeOperation ConditionsProduction (g/L) Productivity (g/L/day)Ref.*G. fujikuroi*Mutant Mor-25Random mutagenesis after UV irradiationSelection based on morphology and pigmentationSmF ^3^Mutants does not accumulate pigmentsSucrose 112.5 g/LBatch fermenter(10-L)8 dayspH 6.828 °C720 mg/L90 mg/L/day[87]*G. fujikuroi*Mutant S109Random mutagenesis increasing the activity of 3-hydroxy-3-methylglutaryl-coenzyme A (HMG-CoA) reductaseSmF ^3^Corn flour 75 g/LRice flour 87.5 g/LAdding glycerol 10 g/L at day 5Batch Bioreactor(2 L in 5 L)28 °C600 rpm0.5 vvm9 days2800 mg/L311.1 mg/L/day[69]*G. fujikuroi*Mutant M27Random mutagenesis of protoplastsScreening during 16 roundsSmF ^3^Corn flour 7.5%Rice flour 8.75%Batch flask (40 mL in 250 mL)28 °C250 rpm7 daysAround 2380 mg/L340 mg/L/day[91]*F. moniliforme*Mutant *γ*-14Random mutagenesis after gamma rays irradiationSelection based on colony morphology and pigmentationSmF ^3^Milk permeate medium and optimization temperature, incubation, pH, inoculum sizeSemicontinuous30 °C150 rpmpH 58 days2400 mg/L300 mg/L/day[88]*F. fujikuroi*Mutant IMI 58289Random mutagenesis of protoplastsIrradiated with Cobalt-60 and Lithium-chloride Genome sequencingSmF ^3^Starch 75 g/LRice flour 87.5 g/LBatch flask (40 mL in 250 mL)28 °C250 g7 days2100 mg/L300 mg/L/day[72]*F. fujikuroi*Mutant NMMultivariate modular metabolic engineering approachGene overexpression of genes involved in GA_3_ biosynthesisSmF ^3^Corn starch 75 g/LRice flour 87.5 g/LBatch flask7 days2890 mg/L412.85 mg/L/day[89]*F. fujikuroi*Mutant CGMCC No: M 2,019,378Regulatory modificationOverexpression of AreA and Lae1 positive regulatorsSmF ^3^Cornstarch, 75; rice four, 87.5; soybean meal, 5; peanut powder, 5 g/LBatch flask28 °C600 rpm7 days3020 mg/L431.43 mg/L/day[90]NM not mentioned. ^3^ submerged fermentation (SmF).

## 9. Market Overview

The agriculture industry has witnessed an upsurge in the utilization of gibberellins due to the escalating need for superior-grade veggies and fruits. Farmers prefer this product due to its ability to enhance plant growth under controlled and predictable conditions [92]. Therefore, agribusiness has needed to implement new strategies; one of the most efficient and modern is the use of GAs as stimulants in the cultivation of various vegetables and fruits, which boosts the demand for the bioproduct.

The gibberellins market is segmented into GA_1_, GA_3_, GA_4_, and GA_7_. The first two are used for growth promotion in crops such as grapes, rice, and wheat. On the other hand, GA_4_ and GA_7_ are mixed in different percentages for commercial presentation, being mainly used in apple and pear crops [93,94,95,96] (Figure 6).

The worldwide market for GAs, which was estimated at USD 907.7 million in 2023, is projected to reach USD 1434.52 million by the end of 2030, with a CAGR of 6.2% [97]. Within the different commercial growth-active GAs, GA_3_ is the main gibberellin produced in commercial industrial-scale fermentations of Gibberella for agronomic, horticultural, and other scientific applications [37]. In addition, this compound accounted for a significant portion of sales in 2021, with a product price ranging between USD 150 and 500 per kilogram [97,98].

The key competitors in this area are countries with impressive economies and technological development, such as China, the United States, Canada, Germany, and Japan [60].

China has over 80 officially registered national companies that specialize in the production of GAs. In 2016, these companies held a dominant market share of 91.06%. The main companies in this field are Zhejiang Qianjiang Biochemical, Shanghai Tongrui Biochemical, Jiangxi Xinruifeng Biochemical, Sichuan Lomon Biochemical, and Jiangsu Fengyuan Biotech. In addition, China has 159 registered gibberellic acid products, of which 102 are single-dose compounds and 57 are mixed compounds. These products have been specifically developed for enhancing crops of grapes, pineapples, cotton, potato, celery, spinach, tomato, cucumber, apple, sugarcane, wheat, corn, and others [99,100]. Currently, companies are not primarily focused on increasing the production of gibberellins, but since these compounds are classified as biochemical pesticides, they are also looking to develop formulations with low toxicity [99]. According to importation data from the United States between August 2023 and February 2024, there was a demand of 122.7 tons from Chinese companies, making this one of the main markets [101]. On the other hand, data from the Beijing Multigrass industry showed a technical capacity of 100 tons in 2018 [102]. These numbers provide an insight into the size of China’s production capacity.

## 10. Conclusions and Future Prospects

Undoubtedly, plant growth regulators are powerful tools for the development of avant-garde agriculture that meets the necessary requirements to satisfy the growing demand for high-quality green products. Specifically, GA_3_ integrates the most important model produced from microbial origin due to its activity in tolerance against abiotic stress, the stimulation of plant development, and the increase in post-harvest life, so the exogenous application of the phytohormone manages to positively intervene in the yield and quality of economically valuable crops. For this reason, it is essential to establish the parameters and methodological strategies that maximize the production of GA_3_ on an industrial scale.

Considering the information presented above, its seems that the production of GA_3_ by solid fermentation exhibits greater promise, yielding higher quantities compared to submerged culture methods. Thus, the exploration of cultivation systems such as drum bioreactors to enhance mass transfer, appears to be a viable strategy for augmenting production yields. Moreover, solid fermentation enables more direct utilization of raw materials such starch, maize flour, wheat bran, rice bran, and citric pulp, among others. On the other hand, the majority of studies have been conducted in submerged fermentations where refined and/or partially solubilized raw materials are common and, thereby, the separation and purification of the GA_3_ is more convenient. However, it is a common fact that in both fermentation techniques, the process time is long, and the conversion of raw materials into product is poor compared to other metabolites produced by fungi. Also, strain selection plays a pivotal role in solid and submerged fermentation processes. Notably, *Fusarium moniliforme* (asexual reproductive stage) has demonstrated superior yields compared to *Fusarium fujikuroi* (sexual reproductive stage), suggesting the potential for genetic mutation approaches to enhance productivity within this strain.

In this respect, the acquired knowledge on GA_3_ regulation and biosynthesis has opened new opportunities for improving regulatory circuits by genetic engineering strategies. Moreover, current methods have enabled directed genetic modifications in the metabolic pathways that result in higher improvements than those obtained by random mutagenesis. Since GA_3_ biosynthesis is an overly complex process, the actual research has explored a global regulatory approach for a multiple-level enhancement that involves both genetic regulation and metabolic engineering with promissory results. Therefore, further studies must be directed toward evaluating the combined effects from the simultaneous manipulation of positive global regulators overexpression, metabolic precursors, and bioprocess conditions.

Finally, the rising demand for gibberellins in agriculture presents both challenges and opportunities for the industry. Meeting market demands efficiently requires the implementation of innovative strategies to optimize production and distribution channels. As gibberellins continue playing a crucial role as growth stimulants in vegetable and fruit cultivation, their significance in modern agricultural practices is expected to grow further. Market segmentation reveals the diverse applications of gibberellin types, with GA_3_ being particularly prominent in large-scale fermentations for agricultural, horticultural, and scientific purposes. The dominance of key players such as China, the United States, Canada, Germany, and Japan underscores the global nature of gibberellin production and trade, presenting opportunities for collaboration and market expansion. China’s significant market share and extensive production capabilities highlight its pivotal role in shaping the future of the gibberellin market. Moreover, as companies shift their focus towards developing formulations with reduced toxicity, there is the potential for innovation and growth in this sector.

## Figures and Tables

**Figure 1 jof-10-00418-f001:**
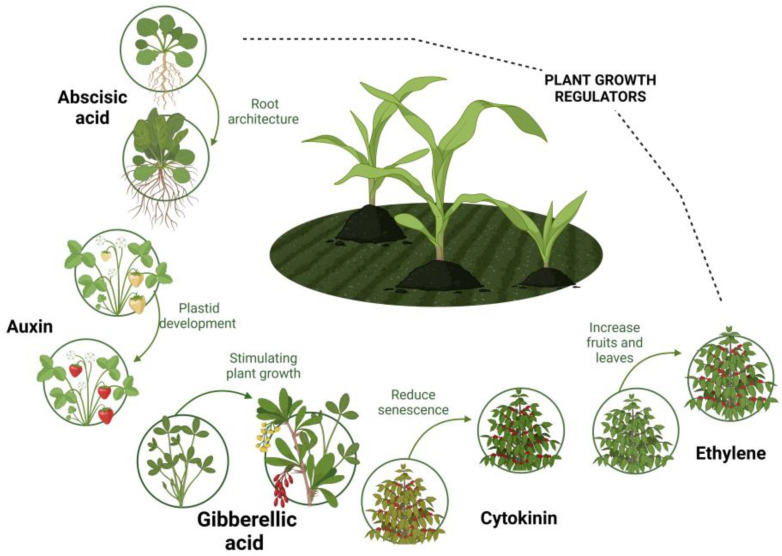
Principal plant growth regulators.

**Figure 2 jof-10-00418-f002:**
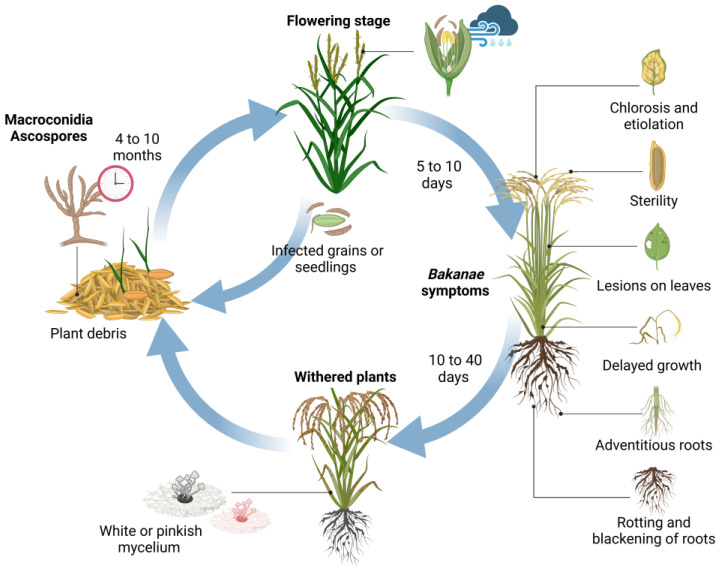
Epidemiology of bakanae disease.

**Figure 3 jof-10-00418-f003:**
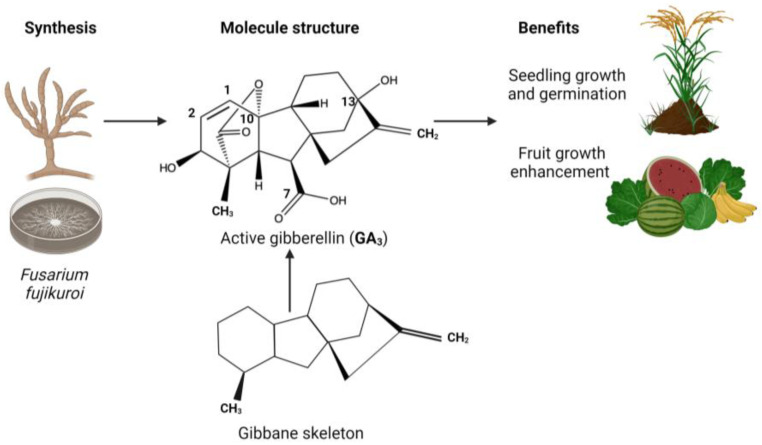
The structure of GA_3_ and its biosynthesis form and benefits.

**Figure 4 jof-10-00418-f004:**
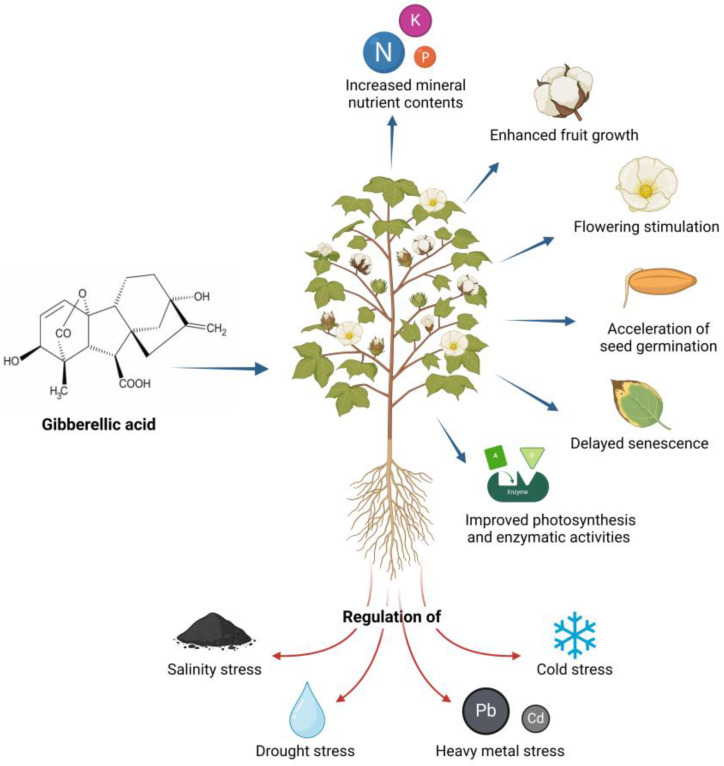
Main biological activities of GA_3_.

**Figure 5 jof-10-00418-f005:**
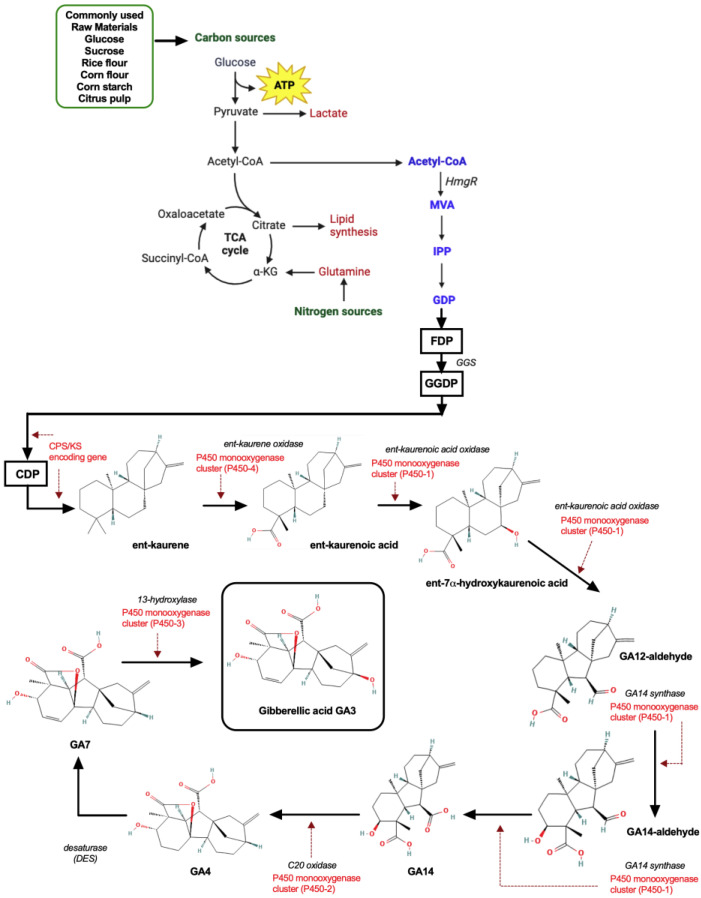
Metabolic pathways involved in GA_3_ biosynthesis from *F. fujikuroi* and raw materials commonly used for GA_3_ production. Regulatory elements are marked in red, and enzymes at each step are marked in italics.

**Figure 6 jof-10-00418-f006:**
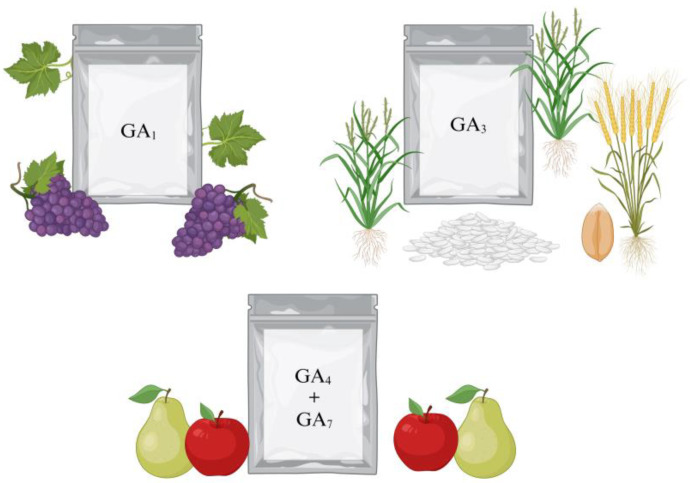
Main growth-active GAs employed in different crops [92].

## Data Availability

Not applicable.

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
