# Peer review of "Bioprocess of Gibberellic Acid by Fusarium fujikuroi: The Challenge of Regulation, Raw Materials, and Product Yields"

_jof, 2024, doi:10.3390/jof10060418_

Round 1

Reviewer 1 Report

Hernández Rodríguez with co-authors prepared a review devoted to production of gibberellic acid by Fusarium fujikuroi and its importance in plant industry. Despite there are numerous review articles on gibberellic acid (GA), the manuscript focuses on improvements of its production processes. Some minor aspects may be considered to make the manuscript more focused and readable. Too much attention was paid to the history and market of GA. Instead, I would propose to include information and discuss GA formulations. Some figures could be edited (2, 3 and 5) or even omitted (7).

Line 78 Re-evaluate and edit the sentence.

Figure 2. Macroconidia – plural form

Figure 3 is too complex for understanding. Please, make it simpler, demonstrate one idea per picture.

Line 202. Edit the sentence.

Figure 5. “defined raw materials” looks strange in the biosynthesis scheme

Two paragraphs beginning since line 369 and 379 need more references

Figure 7 could be deleted because no data are presented

Line 564 enhanced

573-574 check correct species names for fungi, asexusal (Fusarium) and sexual (Gibberella) stages

Author Response

Reviewer 1:

Line 78 Re-evaluate and edit the sentence.

Done, we re-evaluated and edited the sentence.

Page 3, Line 75. Considering GA3 as a promising strategy for the benefit of the agricultural sector.

Figure 2. Macroconidia – plural form

Done, thanks we have changed it.

Page 4, Line 128.

Figure 3 is too complex for understanding. Please, make it simpler, demonstrate one idea per picture.

Done, you’re right. We have modified the figure for a better understanding and the figure caption was improved.

Page 5, Line 186.

Line 202. Edit the sentence.

Done, we have edited the sentence

Page 6, Line 198. Gibberellic acid promotes physiological and metabolic responses in plants.

Figure 5. “defined raw materials” looks strange in the biosynthesis scheme

Done, we have specified “raw materials” in the figure and the figure caption was improved.

Page 9, Line 326.

Two paragraphs beginning since line 369 and 379 need more references

Done, we have modified the sentence and added more references

The strategies mentioned in line 374 are discussed in more detail in Table 1.

Page 10, Line 370-372; 377-380.

Figure 7 could be deleted because no data are presented

Done, you’re right. We have decided to deleted it

Page 17, Line 524.

Line 564 enhanced

Done, thanks, we have changed the sentence for better understanding.

Page 18, Line 555.

573-574 check correct species names for fungi, asexusal (Fusarium) and sexual (Gibberella) stages

Done, thanks, we have corrected the species names

Page 18, Line 563-564. Both are the reproductive stages of the Gibberella pathogen, they were renamed to Fusarium fujikuroi (teleomorph: sexual reproductive stage) and Fusarium moniliforme (anamorph: asexual reproductive stage) and this information was mentioned in the section “2. Bakanae: from the Problem to the Opportunity” (line 100-106). 

Reviewer 2 Report

The authors provided a comprehensive review on the  bioprocess of gibberellic acid, including the origins, plant growth regulations, biosynthetic pathways, and production and related strategies. Moreover, the article was good organized and well written.

1. Usually, the methyls on the nuclei are considered as parts of the skeleton. Please provide the gibbane skeleton with methyls in Figure 3.

2. What did the ladder mean in Figure 7?

Author Response

Reviewer 2

Usually, the methyls on the nuclei are considered as parts of the skeleton. Please provide the gibbane skeleton with methyls in Figure 3.

Done, we have drawn it as you have suggested

Page 5, Line 189.

 What did the ladder mean in Figure 7?

Page 17, Line 524. Done, the staircase represented the increase in gibberellic acid production per country. We have deleted the figure to avoid any confusion.

Reviewer 3 Report

The manuscript by Hernandez Rodriguez et al presents the importance of gibberellic acid, a key plant hormone in the growth and development of plants, which plays a fundamental role in various industrial applications. The manuscript describes this importance since its capacity to regulate a wide range of physiological processes, such as seed germination, stem elongation, flowering and fruit formation. Authors describe its use to increase production and improve the quality of crops, in such a way that the production of fruits and vegetables could be influenced by gibberelic acid, accelerating the growth of crops and promote uniform development, resulting in more abundant and higher quality crops. Authors also described biotechnological improvements using fungi of the Fusarium genus to obtain gibberellic acid, offering a sustainable and efficient alternative to traditional production methods. Through techniques such as microbial fermentation using specific strains of fungi and bacteria, gibberelic acid can be produced on a large scale in a more economical and environmentally friendly way. This biotechnological approaches not only reduces dependence on limited natural resources, such as the plants from which the compound was extracted in the past, but also minimizes waste and contamination associated with conventional methods of chemical synthesis. In summary, the presented manuscript compiles a series of papers in a high quality where gibberellic acid is described as a crucial key in the industry due to its ability to improve productivity, quality and sustainability in a variety of applications, and its biotechnological procurement represents a step forward towards a greener and more efficient future.

I do not have additional comments or suggestions.

Author Response

The reviewer did not have further comments.